# Polymer-Infiltrated Metal–Organic Frameworks for Thin-Film Composite Mixed-Matrix Membranes with High Gas Separation Properties

**DOI:** 10.3390/membranes13030287

**Published:** 2023-02-28

**Authors:** Hyo Jun Min, Min-Bum Kim, Youn-Sang Bae, Praveen K. Thallapally, Jae Hun Lee, Jong Hak Kim

**Affiliations:** 1Department of Chemical and Biomolecular Engineering, Yonsei University, 50 Yonsei-ro, Seodaemun-gu, Seoul 03722, Republic of Korea; 2Pacific Northwest National Laboratory, 902 Battelle Boulevard, Richland, WA 99352, USA; 3Hydrogen Research Department, Korea Institute of Energy Research, 152 Gajeong-ro, Yuseong-gu, Daejeon 34129, Republic of Korea

**Keywords:** gas separation, metal-organic framework, comb copolymer, mixed-matrix membrane, carbon dioxide

## Abstract

Thin-film composite mixed-matrix membranes (TFC-MMMs) have potential applications in practical gas separation processes because of their high permeance (gas flux) and gas selectivity. In this study, we fabricated a high-performance TFC-MMM based on a rubbery comb copolymer, i.e., poly(2-[3-(2H-benzotriazol-2-yl)-4-hydroxyphenyl] ethyl methacrylate)-co-poly(oxyethylene methacrylate) (PBE), and metal–organic framework MOF-808 nanoparticles. The rubbery copolymer penetrates through the pores of MOF-808, thereby tuning the pore size. In addition, the rubbery copolymer forms a defect-free interfacial morphology with polymer-infiltrated MOF-808 nanoparticles. Consequently, TFC-MMMs (thickness = 350 nm) can be successfully prepared even with a high loading of MOF-808. As polymer-infiltrated MOF is incorporated into the polymer matrix, the PBE/MOF-808 membrane exhibits a significantly higher CO_2_ permeance (1069 GPU) and CO_2_/N_2_ selectivity (52.7) than that of the pristine PBE membrane (CO_2_ permeance = 431 GPU and CO_2_/N_2_ selectivity = 36.2). Therefore, the approach considered in this study is suitable for fabricating high-performance thin-film composite membranes via polymer infiltration into MOF pores.

## 1. Introduction

Anthropogenic CO_2_ emissions significantly accelerate global warming; therefore, CO_2_ removal technologies have been extensively investigated [1,2]. In recent years, membrane separation has demonstrated considerable potential as an energy-efficient and environmentally friendly method, particularly for CO_2_ capture from flue gas [3,4]. Polymeric membranes are favorable because of their low cost and high scalability but are generally limited by the trade-off between permeability and selectivity [5]. Mixed-matrix membranes (MMMs), comprising functional porous fillers in the polymer matrix, have emerged as potential candidates for high-performance membranes to enhance the gas separation performance of polymeric membranes [6,7,8].

Metal–organic frameworks (MOFs) are considered next-generation porous materials suitable for gas separation and capture because of their high surface area, tunable porosity, tailorable pore size, and functionality through the combination of various organic linkers and metal clusters [9,10]. Additionally, the ability to control the particle size of MOFs have a great advantage for the MMMs fabrication [10,11]. The interface morphology between the MOF particle and the polymeric chains is critical to the design and fabrication of high-performance MMMs [12]. Insufficient interfacial adhesion of the polymer on the MOF particle creates interfacial voids and considerably decreases the membrane selectivity [6,13]. Additionally, the pore blockage resulting from the polymer penetration should be avoided because gases cannot permeate through the pore-blocked MOFs in the membranes [14,15]. However, the pore size of the MOFs can be tuned with adequate pore blockage. As reported by Ban et al. [16], a zeolitic imidazolate framework (ZIF-8) with enhanced selectivity was obtained by incorporating an ionic liquid into ZIF-8 nanoparticles. Consequently, the effective cage size of ZIF-8 decreased, enabling the size-sieving of CO_2_ against N_2_ or CH_4_. Moreover, partial polymer penetration into the MOF typically facilitates the formation of an intimate interface between the fillers and the polymer matrix, thereby preventing defects between them [17,18].

In terms of the industrial applications of the membranes, the primary challenge involves improving the gas permeance (flux) of the membrane rather than the permeability (material property regardless of the thickness) because a substantial gas flux needs to be processed through the membrane [19,20]. Therefore, thin-film composite (TFC) membranes comprising submicron-thick selective dense layers on highly porous supports have been recently examined for scaling up gas separation membranes [21]. However, for preparing thin-film composite mixed-matrix membranes (TFC-MMMs), the interfacial morphology control between the MOFs and the polymer matrix must be effectively designed because the MOF particle size is relatively large in a submicron-thick selective layer [22].

MOF-808 is a porous framework composed of hexa-zirconium, which exhibits higher adsorption properties for CO_2_ than for other gases, such as N_2_, in addition to sufficient chemical/thermal stability. Plonka et al. [23] investigated the interaction between the MOF-808 structure and CO_2_. The results indicate that CO_2_ that enters the MOF structure is preferentially located inside tetrahedral cages owing to the residual electrons of organic linkers. MOF-808 has two different types of pore sizes with tetrahedral cages (size = 4.8 Å) and adamantine-shaped cages (diameter = 18.4 Å) [24,25]. Although the large pore size (which is higher than the kinetic diameter of N_2_ or CH_4_) of MOF-808 ensures an increased permeability in MMMs, the increase in selectivity is generally limited. Kulak et al. [26] investigated the CO_2_ separation performances of MMMs based on MOF-808 and co-polyimide under different conditions. The permeance increased with the incorporation of MOF-808; however, the separation factor was not considerably different from that of the pristine polymer. Vankelecom group [27,28] reported improved separation factors for MOF-808-based MMMs by the functionalization of MOF-808. The smaller pores (size = 4.8 Å) of MOF-808 can serve as selective pore channels by size-sieving if the pore diameter is tuned and lies between the kinetic diameters of CO_2_ (3.3 Å) and N_2_ (3.64 Å).

In this study, we fabricated high-performance TFC-MMMs based on the polymer-infiltrated MOF-808. The flexible and rubbery copolymer, poly(2-[3-(2H-benzotriazol-2-yl)-4-hydroxyphenyl] ethyl methacrylate)-co-poly(oxyethylene methacrylate) (PBE), was used for the polymer matrix. The rubbery PBE copolymer penetrated into the pores of MOF-808 through a simple blending, which was characterized using various techniques. The PBE infiltrating into the pore of MOF-808 not only improved the close interfacial contact with the polymer matrix but also reduced the effective pore size of PBE/MOF-808 membranes by controlling the pore size of MOF-808, enhancing the CO_2_/N_2_ selectivity. The gas separation performance of the TFC-MMMs was evaluated at 1 atm and 30 °C.

## 2. Materials and Methods

### 2.1. Materials

For synthesizing the copolymer, 2-[3-(2H-Benzotriazol-2-yl)-4-hydroxyphenyl]ethyl methacrylate (BEM, M_w_ = 323.35 g mol^−1^), and poly(oxyethylene methacrylate) (POEM, poly(ethylene glycol) methyl ether methacrylate, M_n_ = 500 g mol^−1^) were obtained from Sigma–Aldrich (St. Louis, MO, USA). Poly(1-(trimethylsilyl-1-propyne)) (PTMSP) was purchased from Gelest Inc. (Morrisville, PA, USA). N,N-Dimethylformamide (DMF), tetrahydrofuran (THF), isopropyl alcohol (IPA), n-hexane, cyclohexane, and methanol were purchased from J.T. Baker (Avantor, Radnor, PA, USA). Azobisisobutyronitrile (AIBN, Mw = 164.21 g mol^−1^). For synthesizing MOF-808, the ZrOCl_2_∙8H_2_O, trimesic acid, DMF, and formic acid were purchased from Acros Organics (Geel, Belgium). All solvents and chemicals were ACS reagent grade and used as received.

### 2.2. Synthesis of MOF-808

MOF-808 powder was synthesized and purified according to the published procedure [25,29]. ZrOCl_2_∙8H_2_O (4.85 g) and trimesic acid (H_3_BTC; 1.05 g) were completely dissolved in DMF/formic acid (225 mL/225 mL), transferred into a 1000 mL screw-capped jar, and sonicated for 30 min. The clear mixed solution was heated at 130 °C for 2 d and subsequently cooled to room temperature. The resulting white powder was collected via filtration and washed thrice with 150 mL of DMF. The sample was then immersed in 100 mL of DMF for 3 d, replacing fresh DMF daily. The DMF-immersed resulting powder was filtered under vacuum and soaked in 100 mL of acetone for 3 d with replacing fresh acetone daily. The final purified product was collected via centrifugation and subsequently evacuated at room temperature for 24 h.

### 2.3. Synthesis of the PBE Comb Copolymer

The PBE comb copolymer was synthesized via free-radical polymerization, according to our previously reported procedure [30]. Initially, 2 g of BEM was dissolved in 30 mL of DMF at room temperature for 1 h. Subsequently, 8 g of POEM and 0.002 g of AIBN were added to the solution, which was purged with N_2_ gas for 30 min. The polymerization reaction was performed at 90 °C for 20 h. The resultant polymer solution was precipitated in an n-hexane/IPA mixture (8:2). Precipitation was performed thrice to purify the polymer, with the resultant rubbery polymer dried in a vacuum oven to completely remove the residual solvent.

### 2.4. Preparation of TFC-MMMs

Various amounts of MOF-808 powders (0.02, 0.04, 0.06, 0.08, and 0.1 g) were dispersed in 0.2 g of MeOH solution under vigorous stirring and sonication for 1 h, respectively. Simultaneously, 0.2 g of PBE copolymer was dissolved in 1 g of MeOH under vigorous stirring at room temperature. Subsequently, the MOF-808 solution was added to the polymer solution and stirred for 24 h until the solution was well-dispersed. To prevent the selective polymer solution from penetrating the porous support, an RK control coater (Control RK Print-Coat Instruments Ltd., UK) was used to coat 0.5 wt% of PTMSP/cyclohexane solution on the microporous polysulfone (Psf) support layer as a gutter layer. After drying for 3 h, the PBE/MOF-808 solution was directly coated on the PTMSP/Psf membranes. The resulting membranes, with different ratios of MOF-808 particles, were labeled as 10%, 20%, 30%, 40%, and 50%. The prepared membranes were then dried overnight at room temperature and consecutively vacuum-dried overnight at 80 °C.

### 2.5. Characterization

The Brunauer–Emmett–Teller (BET) surface area from N_2_ adsorption–desorption isotherms at 77 K of MOF-808 and PBE@MOF-808 were determined using a 3Flex (Micromeritics Instruments, Norcross, GA, USA). Before the N_2_ adsorption–desorption measurement, approximately 100 mg of the samples were degassed under vacuum at 150 °C for 24 h. The chemical interactions between PBE copolymers and MOF-808 membranes were investigated using Fourier transform infrared spectroscopy (FT-IR). The morphologies of the membranes and particles were examined using a field-emission scanning electron microscope (FE-SEM, JSM-6701F, JEOL Ltd., Tokyo, Japan) and transmission electron microscope (TEM, Libra 120, Zeiss, operated at 120 kV). The thermal stability of the materials was investigated using thermogravimetric analysis (TGA) through a DTA/TGA analyzer (TA instruments, New Castle, DE, USA) at a heating rate of 10 °C/min under a nitrogen environment. The chain morphology of the polymers was analyzed using differential scanning calorimetry (DSC, (DSC8000, Perkin Elmer, Waltham, MA, USA)) at a scan rate of 10 °C/min under a nitrogen atmosphere. The crystalline structure of the membranes was characterized using X-ray diffraction (XRD, RINT2000 wide-angle goniometer, Rigaku, Tokyo, Japan) with a Cu cathode operated at 40 kV and 300 mA.

### 2.6. Gas Permeation Measurements

The pure gas permeance and separation properties were determined using a constant pressure/variable volume apparatus (Airrane Co., Ltd., Cheongju, Korea), according to our previously reported method [22]. The area of the membrane was approximately 10.2 cm^2^. All experiments were carried out at 30 °C, 1 bar. Each membrane was tested five times, and the average estimated error in gas permeance was approximately 5%. The gas permeance of the membrane was expressed in gas permeation units (GPU) (1 GPU = 10^−6^ cm^3^ (STP) cm^−2^ s^−1^ cmHg^−1^).

## 3. Results

### 3.1. Characterization of MOF-808 Nanoparticles and PBE/MOF-808 MMMs

The PBE comb copolymer consists of rigid hydrophobic BEM chains and rubbery hydrophilic POEM chains (Figure 1a). We found that the PBE copolymer is suitable as a thin-film composite membrane for the CO_2_ separation [30]. The rubbery POEM groups are known to have excellent CO_2_ selectivity owing to the dipole–quadrupole interaction and a high CO_2_ permeance, owing to the amorphous state of the polymer chains, whereas the rigid PBEM chains can inhibit N_2_ permeance and membrane plasticization. The rubbery state of the PBE copolymer can be easily penetrated into the pores of MOF-808 and improve the interfacial morphology between the PBE copolymer and MOF-808 [18]. Particularly, the PBE copolymer is alcohol-soluble, which facilitates the fabrication of TFC membranes because the solution does not dissolve the porous polymer support substrates such as Psf [31].

We synthesized the MOF-808 nanoparticles, and octahedral particles of sizes 500–600 nm were observed from the SEM and TEM images (Figure 1a,b). Appendix A presented the N_2_ isotherm at 77 K and the pore size distribution for the degassed MOF-808, respectively. Based on the N_2_ isotherm, the BET surface area of MOF-808 was calculated as 2070 m^2^ g^−1^; the high surface area of MOF-808 indicates that it is sufficient to improve the permeability of MMMs by incorporating polymer into MOF-808. Using a density functional theory method, the pore size distributions were evaluated as 7.3 and 18.6 Å; the smaller pore was slightly different from the theoretical data (4.8 Å). The obtained results are consistent with those in the literature, indicating the successful synthesis of the MOF particles [32,33]. Moreover, the high CO_2_ adsorption properties and CO_2_ affinity of MOF-808 nanoparticles were confirmed (Appendix A).

The FT-IR spectra of MOF-808, PBE comb copolymer, and PBE/MOF-808 MMMs with various MOF-808 contents were obtained (Figure 2a) to investigate the chemical interaction between MOF-808 filler and the PBE copolymer matrix. Pristine PBE copolymer has strong absorption bands at 1726 and 1100 cm^−1^, assigned to the stretching vibrations of carbonyl (C=O) and ether (C–O–C) functional groups, respectively [34]. PBE/MOF-808 MMMs maintained the original absorption band positions of PBE copolymer after incorporation with MOF-808, revealing that only a weak chemical interaction occurs between the polymer matrix and fillers. Strong chemical interactions between polymer matrix and fillers can densify and rigidify polymer chains and adversely affect the gas permeation [15].

The crystalline structures of the PBE copolymer, MOF-808, and PBE/MOF-808 MMMs were characterized using XRD patterns (Figure 2b). All the samples clearly showed isostructural to the pristine MOF-808. The PBE copolymer exhibits a broad peak at approximately 20.0° without a sharp crystalline peak, implying that the polymer structure is amorphous. In contrast to other poly(ethylene oxide) (PEO)-based polymers, the amorphous nature of the PBE copolymer is observed owing to the short chains of ethylene oxide units in the copolymer. The sharp crystalline peaks of MOF-808 were maintained after incorporation with the PBE polymer matrix, and their intensities gradually increased with the increasing MOF-808 content. As corroborated by the FT-IR spectra, the PBE copolymer matrix and MOF-808 filler do not sufficiently interact to affect the crystalline structure of MOF-808.

The chain morphologies of the PBE copolymer and PBE/MOF-808 MMMs were investigated through DSC analysis (Figure 2c). The PBE copolymer exhibits a glass transition temperature (T_g_) at −53.7 °C. No sharp endothermic peaks attributed to polymer melting are observed, thus demonstrating the crystalline-free amorphous nature of the PBE copolymer, consistent with the XRD results. In terms of gas permeation, an amorphous structure is considered desirable because hard crystallite in a polymer matrix can act as a barrier to gas molecules and decrease gas permeability. T_g_ of PBE/MOF-808 MMMs gradually increases from −52.7 °C to −47 °C with the increasing MOF-808 content. The increase in T_g_ is generally induced by suppressing the polymer chain mobility and flexibility. As confirmed by the FT-IR spectra, only weak chemical interactions are observed between the PBE copolymer matrix and MOF-808 fillers. Thus, the change in T_g_ possibly originates from the physical penetration of polymer chains into the MOF-808 structure. The polymer chains could penetrate the pores of the MOF-808 structure and exhibit moderate flexibility and mobility.

We investigated the surface area and pore size of washed MOF-808 nanoparticles (Figure 3) to further verify the pore infiltration of polymer chains. The washed MOF-808 particles were prepared as follows: first, MOF-808 particles were blended with PBE solution, similar to the preparation of PBE/MOF-808 MMMs; then, MOF-808 particles were collected from the solution via centrifugation. The resultant particles were washed with simultaneous sonication and centrifuged thrice with MeOH three times to remove the residual polymer on the surface of the MOF-808 particles. Not only the surface area of the MOF-808 mixed with the PBE copolymer decreased from 2070 to 696 m^2^ g^−1^ but also the pore volume decreased. Particularly, two peaks at 7.3 and 18.6 Å, corresponding to the small and large pores, respectively, are observed for pristine MOF-808. However, MOF-808 mixed with the PBE copolymer presents peaks from the pore size distribution at approximately 3.9 and 15.9 Å, which are lower than those of the pristine MOF. These results imply the penetration of the polymer chain into the pores of MOF-808. The PBE copolymer chains are entangled in the pores of MOF-808 filler; therefore, they can adhere to the MOF structure even after extensive washing with polymer-dissolving solvents. Considering that the theoretical pore sizes of MOF-808 are 4.8 and 18.4 Å, the effective pore size of the polymer-infiltrated MOF would be lower than the obtained data [32]. The pore-tuned MOF-808 can act as a CO_2_-philic nanocage in the membrane matrix, which selectively facilitates gas molecules. Furthermore, the PBE@MOF-808 sample exhibits an improved CO_2_ adsorption property compared to the pristine MOF-808. This is a result of the quadrupole-dipole interaction of the CO_2_ molecule with the ethylene oxide group of the PBE chain and shows that the PBE chain is well entangled through the pores of MOF-808. Furthermore, the increase in adsorption is related to the decreased pore size of the MOF-808. The adsorption strength can be increased because the target molecule can interact with the other walls of the pore without being affected by only one part of the pore [35]. Therefore, the membrane performance can be improved by incorporating the polymer-infiltrated MOF-808 into the polymer matrix [36].

We investigated the thermal stability of the PBE comb copolymer, MOF-808 particles, and PBE/MOF-808 MMMs through TGA (Appendix A). The pristine PBE copolymer demonstrates excellent thermal stability up to 200 °C. For MOF-808 particles, 0–200 °C is associated with the volatility of the solvent contained in the adsorbent. Generally, the solvent trapped in MOF-808 nanoparticles is evaporated when the temperature increases and the weight is partially decreased. The weight loss in the range of 200–400 °C occurs because the formic acid in MOF-808 is removed. Finally, the weight loss occurring between 400 and 600 °C is considered thermal decomposition owing to the loss of the MOF-808 linker. After incorporating MOF-808 particles, all membranes demonstrate sufficient thermal stability to be applied as CO_2_ separation membranes under post-combustion process conditions [37].

### 3.2. Preparation and Gas Separation Performance of TFC-MMMs

Thin-film structures are highly desirable for practical applications in membrane processes because the gas flux is one of the most critical factors in the process. Hence, the thickness of the selective layer should be minimized to maximize the permeance (flux) of the membranes. However, due to the thin selective layer, reducing defects in TFC-MMMs is much more difficult than in thick and dense membranes. We fabricated TFC-MMMs via a simple bar-coating method on a porous Psf support. The morphology of PBE/MOF-808 TFC-MMMs was confirmed through cross-sectional SEM images, with the mixture of PBE copolymer and MOF-808 particles adequately coated on the Psf supporting layer without penetration into the porous support (Figure 4a). MOF-808 particles were well-dispersed on the Psf supporting layer and encapsulated by the PBE comb copolymer. The PBE copolymer acts as a mechanically strong matrix to form a thin selective layer and as a binder between MOF-808 particles and the Psf supporting layer to prevent structural defects. Moreover, the boundary between the MOF-808 particles and the PBE polymer layer is uncertain, implying their considerable compatibility (Figure 4b). The defects from the fabrication of TFC-MMMs are not observed because of the intimate contact between the polymer matrix and the polymer-infiltrated MOF-808 nanoparticles. Moreover, although the selective layer thickness (approximately 350 nm) is less than the MOF-808 nanoparticle size, the polymer covers the surface of the nanofillers without defects. As shown in Figure 4c, partial penetration of polymer chains into the MOF-808 particles enhances interfacial contact, enabling the successful fabrication of TFC-MMMs despite the high loading of MOF particles in the membranes.

The pure gas permeance of membranes was characterized using a constant pressure/variable volume apparatus at 30 °C. The CO_2_ and N_2_ permeance and CO_2_/N_2_ selectivity of PBE/MOF-808 TFC-MMM with various MOF-808 contents are presented in Figure 5 and Table 1. As the PBE copolymer contains CO_2_-philic polar groups, such as ether oxygen in POEM chains and tertiary amine groups in triazole in its structure, the pristine PBE membrane exhibits moderate gas separation performance (CO_2_ permeance = 431 GPU and CO_2_/N_2_ selectivity = 36.2). With the increasing MOF-808 content in the TFC-MMMs, both CO_2_ permeance and CO_2_/N_2_ selectivity of the membranes are rapidly enhanced. The PBE/MOF-808 40% membrane presents the optimal gas separation performance (CO_2_ permeance = 1069 GPU and CO_2_/N_2_ selectivity = 52.7). The PBE/MOF-808 40% membrane exhibits a 2.5-fold increase in CO_2_ permeance and a 45% increase in CO_2_/N_2_ selectivity compared with that of the pristine PBE membrane. Both CO_2_ and N_2_ permeance increased with the incorporation of MOF-808. The significant increase in permeance of the TFC-MMMs indicates that the gases effectively diffuse through the large pores of the MOF-808 (adamantine-shaped cages with the size of 15.6 Å), and the polymer impregnation does not result in pore blockage or dead MOF pores. Although pristine MOF-808 is known to have an exceedingly large pore size that inhibits the size-sieving of the CO_2_ molecules, the CO_2_-philic polymer-encapsulated MOF-808 particles serve as selective nanocages for CO_2_ molecules. The high CO_2_-affinity of the nanoparticles can accelerate the CO_2_ transport relative to that of N_2_ into the MOF-808 nanoparticle pores. Moreover, pores partially covered with the polymer matrix can selectively sieve the gas molecules. The cumulative effect of the polymer-infiltrated MOF-808 considerably enhances the gas selectivity of the membranes. However, both permeance and selectivity are decreased when MOF-808 is added in excess of 50%. The excess MOF-808 particles are not uniformly dispersed in the polymer matrix and are aggregated by themselves, as shown in the surface SEM image of the membranes (Appendix A). Due to the aggregation, not all incorporated MOF particles are activated to function as selective diffusion pathways for gas transport. As a result, both permeability and selectivity decrease, similar to the membranes with low loading of MOF-808. A MOF-808 membrane without the PBE copolymer was also prepared on a PTMSP-coated Psf support using the same method. The neat MOF-808 membrane showed poor gas separation performance with a CO_2_ permeance of 12,000 GPU and CO_2_/N_2_ selectivity of 1.9, indicating the importance of the PBE copolymer matrix to minimize structural defects.

The gas separation performance of the PBE/MOF-808 membranes is compared with those of other TFC-MMMs reported in the literature (Figure 6 and Table 1) [38,39,40,41,42,43]. The gas separation performance of the TFC-MMMs in our study demonstrates high selectivity owing to the selective nanochannels originating from polymer-entrapped MOFs. Moreover, the permeance of the PBE/MOF-808 40% membrane satisfies the target performance required for membranes to be used in practical post-combustion processes (CO_2_ permeance >1000 GPU and CO_2_/N_2_ selectivity >30) [44]. Using a rubbery and flexible PBE comb copolymer, polymer-infiltrated MOF-808 nanoparticles can be prepared through simple blending. Polymer infiltration into the MOF particles induces intimate contact with the polymer matrix, enabling the successful fabrication of thin-film membranes and the formation of effective nanochannels to selectively transport CO_2_. Thus, an efficient approach to fabricating high-performance thin-film gas separation membranes is presented.

## 4. Conclusions

In this study, we fabricated high-performance thin-film membranes by incorporating MOF-808 into the PBE comb copolymer. The flexible and rubbery PBE copolymer was used for the matrix of the membrane, which has moderate gas separation performance and excellent film-forming ability. The flexible PBE chain was partially infiltrated into the pores of MOF-808 through a simple mixing method, and the infiltrated PBE increased the gas separation selectivity by improving the interfacial contact with the polymer matrix and reducing the effective pores of MOF-808. Owing to the intimate contact between the MOF nanoparticles and the PBE polymer matrix, defect-free TFC-MMMs (thickness = 350 nm) were successfully fabricated with a high loading of MOF-808 nanoparticles. The penetration of the polymer matrix could remarkably improve the selectivity and interfacial compatibility of the MOFs. When the MOF-808 content in the membrane was increased up to 40%, the permeance and selectivity continuously increased, and PBE/MOF-808 40% membrane demonstrated the optimal performance with a CO_2_ permeance of 1069 GPU and CO_2_/N_2_ selectivity of 52.7, considerably higher than that of the pristine PBE membrane (CO_2_ permeance = 431 GPU and CO_2_/N_2_ selectivity = 36.2). The resultant performance satisfies the criteria for practical applications of CO_2_ capture in post-combustion.

## Figures and Tables

**Figure 1 membranes-13-00287-f001:**
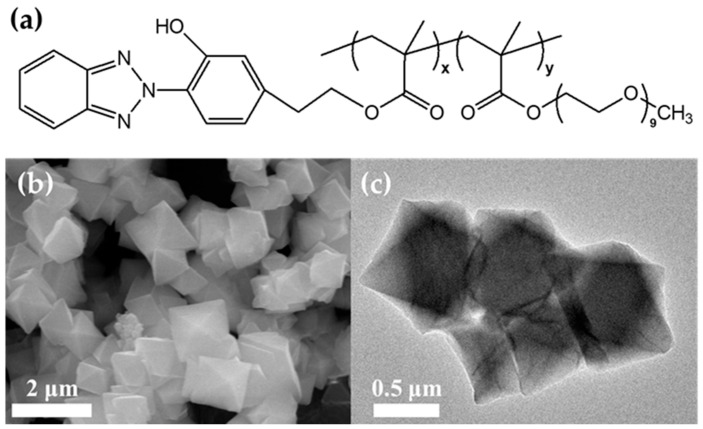
(**a**) Chemical structure of the PBE comb copolymer. (**b**) SEM and (**c**) TEM images of the synthesized MOF-808 nanoparticles.

**Figure 2 membranes-13-00287-f002:**
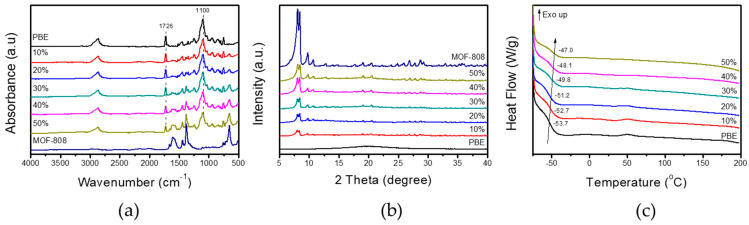
(**a**) FT-IR spectra and (**b**) XRD patterns of the PBE comb copolymer, MOF-808 nanoparticles, and PBE/MOF MMMs with different ratios. (**c**) DSC graphs of the PBE copolymer and PBE/MOF MMMs.

**Figure 3 membranes-13-00287-f003:**
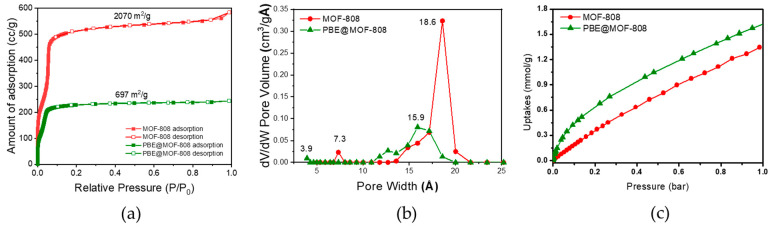
(**a**) BET isotherms. (**b**) pore size distributions. (**c**) CO_2_ adsorption isotherm of pristine MOF-808 and washed MOF-808 after being mixed with PBE copolymer (PBE@MOF-808).

**Figure 4 membranes-13-00287-f004:**
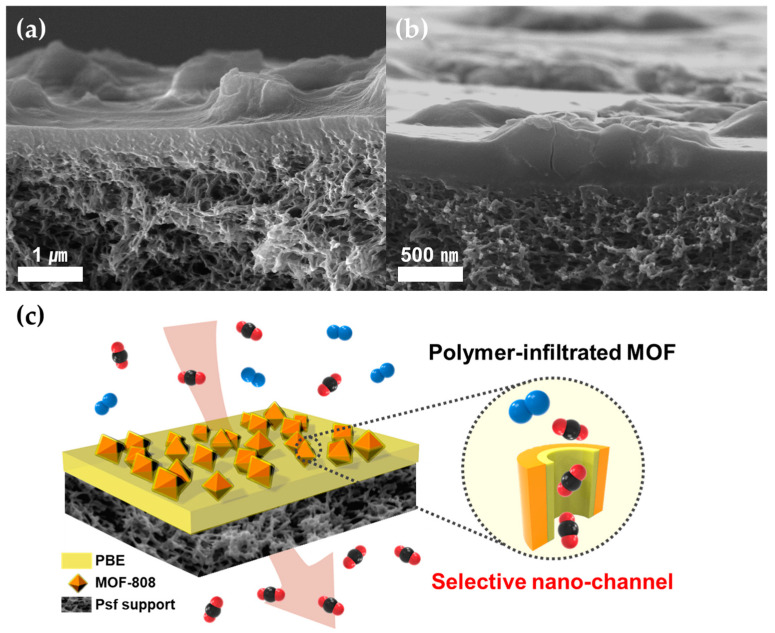
(**a**,**b**) Cross-sectional SEM images of PBE/MOF-808 40% TFC-MMM. (**c**) Schematic of TFC-MMM based on PBE-infiltrated MOF-808.

**Figure 5 membranes-13-00287-f005:**
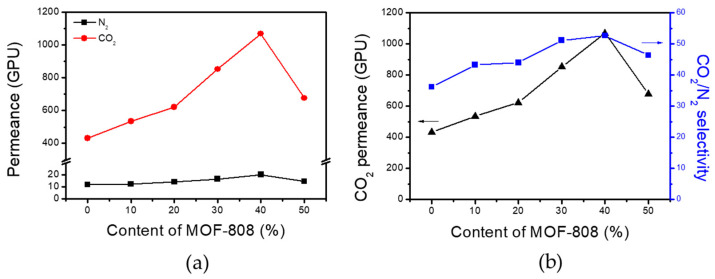
(**a**) CO_2_ and N_2_ permeance of TFC-MMMs with varying MOF-808 content. (**b**) CO_2_ permeance and CO_2_/N_2_ selectivity of TFC-MMMs with increasing MOF-808 content.

**Figure 6 membranes-13-00287-f006:**
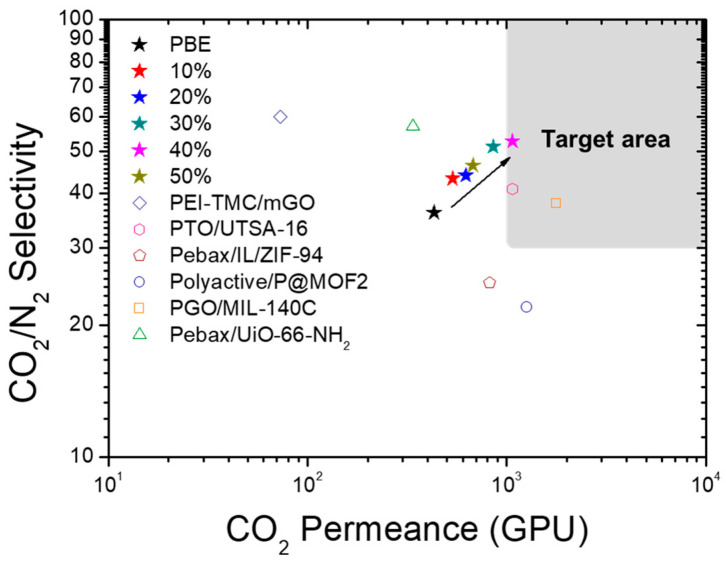
CO_2_ permeance versus CO_2_/N_2_ selectivity for PBE and PBE/MOF-808 membranes and other TFC-MMMs reported in the literature (details in Table 1).

**Table 1 membranes-13-00287-t001:** CO_2_ separation performance of PBE/MOF-808 membranes and previously reported TFC-MMMs.

Sample	CO_2_ Permeance (GPU)	CO_2_/N_2_ Selectivity	Condition (T (°C)/P (bar))	Reference
PBE	431	36.2	30 °C, 1 bar	This study
MOF-808 10%	535	43.3		
MOF-808 20%	623	44.0		
MOF-808 30%	853	51.2		
MOF-808 40%	1069.0	52.7		
MOF-808 50%	677	46.4		
PEI-TMC/mGO	73	60	25 °C, 0.25 bar	[38]
PTO/UTSA-16	1070	41.0	30 °C, 1 bar	[39]
Pebax/IL/ZIF-94	819	25	35 °C, 3 bar	[40]
Polyactive/P@MOF2	1260	22	35 °C, 3 bar	[41]
PGO/MIL-140C	1768	38	30 °C, 1 bar	[42]
Pebax/UiO-66-NH_2_	338	57	25 °C, 2 bar	[43]

## Data Availability

Not applicable.

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
