# Peer review of "Polymer-Infiltrated Metal–Organic Frameworks for Thin-Film Composite Mixed-Matrix Membranes with High Gas Separation Properties"

_membranes, 2023, doi:10.3390/membranes13030287_

Round 1
Reviewer 1 Report
The paper is of interest for the membrane community and there are only few points to be addressed.
1) The BET surface of the MOF-808 seems a bit too high in comparison on what is possible to find in literature for the same MOF. Why?
2) The BET calculations performed on the MMMs, do take into account the different amount of MOF and polymer?
3) Why there is a decrease in both Permeance and selectivity at 50% loading? If the reason are defects, the Permeance should increase.
4) Figure 6, accordingly to Merkel 2010, the Target area at 1000GPU for CO2 starts at selectivity 30 and not 20.
5) If the polymer chains infiltrates in the pore of the MOF, the P should decrease do to pore blocking. Why here it should increase the Permeance?
Author Response
Reviewer #1
Comments to the Author
The paper is of interest for the membrane community and there are only few points to be addressed.
Comments #1
The BET surface of the MOF-808 seems a bit too high in comparison on what is possible to find in literature for the same MOF. Why?
Response #1
As suggested, the BET surface of the MOF-808 synthesized in this study is high. However, there was no significant difference in the surface area of MOF-808 between our study and other studies cited in our paper (2060 m2g-1). For some papers, the surface area of the MOF-808 (2424 m2g-1) was much larger than our value [Karmakar et al, Covalent grafting of molecular photosensitizer and catalyst on MOF-808: effect of pore confinement toward visible light-driven CO2 reduction in water. Energy Environ. Sci. 2021, 14, 2429-2440; Peng et al, A versatile MOF-based trap for heavy metal ion capture and dispersion. Nature communications 2018, 9, 187, doi:10.1038/s41467-017-02600-2]. The surface area of MOF can vary within a batch type or condition even using the same synthesis method. The BET surface area of ​​MOF-808 synthesized in this study is believed to be the value seen in other papers.
Comments #2
The BET calculations performed on the MMMs, do take into account the different amount of MOF and polymer?
Response #2
The ratio between polymer and MOF did not significantly affect the BET or pore-size distribution because excess polymer mixed with MOF was completely removed during the washing process. Because the amount of polymer that penetrates the MOF is very small, mixing different amounts of polymer with the MOF has little difference in the degree of penetration.
Comments #3
Why there is a decrease in both Permeance and selectivity at 50% loading? If the reason are defects, the Permeance should increase.
Response #3
As suggested, the permeance may be increased by defects between the polymer and the MOF particles. However, the main cause of the decrease in permeance is the aggregation of MOF particles. When a large number of MOF particles are added, they are not uniformly distributed in the polymer matrix and aggregate with each other. Therefore, diffusion pathways are not well formed in polymer matrices. Because the pores of the agglomerated MOF particles were not activated due to aggregation, a selective diffusion path for gas transport was not provided. As a result, similar results were obtained with TFC-MMM with low MOF-808 loading, with both permeance and selectivity reduced.
On page 9, we added the following descriptions.
“Due to the aggregation, not all incorporated MOF particles are activated to function as selective diffusion pathways for gas transport. As a result, both permeability and selectivity decrease, similar to the membranes with low loading of MOF-808.”
Comments #4
Figure 6, accordingly to Merkel 2010, the Target area at 1000GPU for CO2 starts at selectivity 30 and not 20.
Response #4
As suggested, we revised Figure 6. Thank you for your comments.
The shaded target area in Figure 6 represents the membrane properties required for commercial CO2/N2 separation, referenced in Figure 8 of Merkel's 2010 paper. According to Merkel's paper, the membranes with a selectivity of less than 20 are not suitable for the separation process regardless of their permeance. Also, the capture cost is strongly dependent on the membrane selectivity when the selectivity is less than 30.
Figure 6. CO2 permeance versus CO2/N2 selectivity for PBE and PBE/MOF-808 membranes and other TFC-MMMs reported in the literature (details in Table 1).
Comments #5
If the polymer chains infiltrates in the pore of the MOF, the P should decrease do to pore blocking. Why here it should increase the Permeance?
Response #5
The increase in membrane permeace with the incorporation of MOF-808 is due to the large pore of the MOF-808. As described in the paper, the MOF-808 has two kinds of pores; tetrahedral cages (size = 4.8 Å) and adamantine-shaped cages (diameter = 18.4 Å). As confirmed by the pore-size distribution data of polymer-infiltrated MOF-808, the sharp peak at 15.6 Å can be observed, even after the infiltration of the polymer. Noting that the kinetic diameters of the target gases are much smaller than the pore size of MOF-808, the results show that the gases can directly diffuse through the large pore of the MOF-808. Therefore, the permeance of the membranes is improved with increasing MOF-808. Thank you for your comments.
The relevant explanations are described on page 8, line 303.
“Both CO2 and N2 permeance increased with the incorporation of MOF-808. The significant increase in permeance of the TFC-MMMs indicates that the gases effectively diffuse through the large pores of the MOF-808 (adamantine-shaped cages with the size of 15.6 Å), and the polymer impregnation does not result in pore blockage or dead MOF pores.”

Reviewer 2 Report
The authors synthesized the PBE/MOF-808 membranes and investigated their performance on selective CO2/N2 adsorption. The manuscript can be accepted after some revisions.
1. It is difficult to clarify the result of rubbery copolymer penetrating through the pores of MOF-808, because the pore is quite small (1.86 nm) and cannot found from the SEM images in Fig. 4. Further evidence should be provided.
2. The original MOF-808 should also be taken to evaluate the CO2 permeance and CO2/N2 selectivity.
Author Response
Reviewer #2
Comments to the Author
The authors synthesized the PBE/MOF-808 membranes and investigated their performance on selective CO2/N2 adsorption. The manuscript can be accepted after some revisions.
Comments #1
It is difficult to clarify the result of rubbery copolymer penetrating through the pores of MOF-808, because the pore is quite small (1.86 nm) and cannot found from the SEM images in Fig. 4. Further evidence should be provided.
Response #1
As suggested, it is difficult to directly confirm the polymer penetration into the MOF-808. Although the pore of the MOF is small, the polymers can partially penetrate through the pores of the MOF-808, because the polymer is fully dissolved in the solvent, and can freely move around the MOF. Duan et al. confirmed the penetration of crystalline high-molecular weight PEO (900,000 g mol-1) into UiO-66 particle (pore diameter = 0.6 nm) through solid-state NMR spectroscopy. The observation matched with the result of decrease in surface area [Duan et al, Polymer infiltration into metal–organic frameworks in mixed-matrix membranes detected in situ by NMR. J. Am. Chem. Soc. 2019, 141, 7589-7595, doi:10.1021/jacs.9b02789.]. Our PBE copolymer is more flexible due to its amorphous nature and has a lower molecular weight than crystalline PEO, allowing it to penetrate MOFs more easily. It should also be noted that the PBE copolymer is partially entangled with MOF-808, which does not mean that the entire polymer chain completely penetrates into the pores and blocks them.
Although it is hard to find direct evidence of polymer infiltration due to small scale, the MOF-808 mixed with the polymer after the thorough removal of the surface-coated polymer (PBE@MOF-808) clearly shows the decrease in surface area and pore volume compared to the pristine MOF. Furthermore, the PBE@MOF-808 sample shows an improved CO2 adsorption compared to the pristine MOF-808. This is a result of the quadrupole-dipole interaction of CO2 with the ethylene oxide group of the PBE chain, and shows that the PBE chain is well entangled through the pores of MOF-808.
We revised Figure 3 as follows.
Figure 3. (a) BET isotherms, (b) pore size distribution, (c) CO2 adsorption isotherm of pristine MOF-808, and washed MOF-808 after mixed with PBE copolymer (PBE@MOF-808).
On page 6, line 244, we added the following descriptions.
“Furthermore, the PBE@MOF-808 sample exhibits an improved CO2 adsorption property compared to the pristine MOF-808. This is a result of the quadrupole-dipole interaction of CO2 molecule with the ethylene oxide group of the PBE chain, and shows that the PBE chain is well entangled through the pores of MOF-808. Also, the increase in adsorption is related to the decreased pore size of the MOF-808. The adsorption strength can be increased because the target molecule can interact with the other walls of the pore without being affected by only one part of the pore [35]. Therefore, the membrane performance can be improved by incorporating the polymer-infiltrated MOF-808 into the polymer matrix [36].”
Comments #2
The original MOF-808 should also be taken to evaluate the CO2 permeance and CO2/N2 selectivity.
Response #2
As suggested, a better way to analyze the effect of MOF on gas separation is to obtain the gas separation performance of the pristine MOF membrane. Unfortunately, however, it is difficult to fabricate pristine defect-free MOF membranes on the Psf support without polymer binder because MOFs are highly crystalline and rigid. It is very challenging to eliminate the defects between the MOF particles especially on a flexible polymer substrate. As a result, MOF membranes in polymer-based thin film composites are generally defective and exhibit very low selectivity. Considering the deterioration of gas separation performance upon excessive MOF loading, the higher the proportion of MOF particles, the lower the gas separation performance. Assuming an ideal MMM, the Maxwell model can be used to theoretically extrapolate the gas separation performance of pristine MOF-808, but the application of the model to thin film composite membranes may be inappropriate and may therefore have a large margin of error.

Reviewer 3 Report
The authors report the infiltration of the polymer poly(2-[3-(2H-benzotriazol-2-yl)-4-hydroxyphenyl] ethyl methacrylate)-co-poly(oxyethylene 16 methacrylate) (PBE) in a porous MOF-808 and the preparation of a polymer membrane for the CO2 adsorption. The work presented is sound, with the characterization performed supporting the conclusions reported. For this reason, I believe the work should be accepted for publication, after some minor revisions. My principal questions regard the adsorption studies:
How does the CO2 adsorption of the polymer-infiltrated MOF particles compare with the MOF 808 itself? Does the polymer infiltration actually improve the CO2 adsorption of the MOF?
How does the MOF itself compares to the polymer/membrane? Does the MOF have the same selectivity?
If the MOF was deposited as a film in the substrate how would it compare with the mixed-matrix described in this work?
Author Response
Reviewer #3
Comments to the Author
The authors report the infiltration of the polymer poly(2-[3-(2H-benzotriazol-2-yl)-4-hydroxyphenyl] ethyl methacrylate)-co-poly(oxyethylene 16 methacrylate) (PBE) in a porous MOF-808 and the preparation of a polymer membrane for the CO2 adsorption. The work presented is sound, with the characterization performed supporting the conclusions reported. For this reason, I believe the work should be accepted for publication, after some minor revisions. My principal questions regard the adsorption studies:
Comments #1
How does the CO2 adsorption of the polymer-infiltrated MOF particles compare with the MOF 808 itself? Does the polymer infiltration actually improve the CO2 adsorption of the MOF?
Response #1
As suggested, we investigated the CO2 adsorption isotherm of the polymer-infiltrated MOF-808 (PBE@MOF-808). As shown in Figure 3c, the PBE@MOF-808 sample exhibits an improved CO2 adsorption property compared to the pristine MOF-808. This is a result of the quadrupole-dipole interaction of CO2 molecule with the ethylene oxide group of the PBE chain, and shows that the PBE chain is well entangled through the pores of MOF-808. Also, the increase in adsorption is related to the decreased pore size of the MOF-808. The adsorption strength can be increased because the target molecule can interact with the other walls of the pore without being affected by only one part of the pore. [Kim et al, High SF6/N2 selectivity in a hydrothermally stable zirconium-based metal–organic framework. Chem. Eng. J. 2015, 276, 315-321, doi:10.1016/j.cej.2015.04.087.].
Figure 3c. CO2 adsorption isotherm of pristine MOF-808 and the polymer-infiltrated MOF-808.
On page 6, line 244, we added the following descriptions.
“Furthermore, the PBE@MOF-808 sample exhibits an improved CO2 adsorption property compared to the pristine MOF-808. This is a result of the quadrupole-dipole interaction of CO2 molecule with the ethylene oxide group of the PBE chain, and shows that the PBE chain is well entangled through the pores of MOF-808. Also, the increase in adsorption is related to the decreased pore size of the MOF-808. The adsorption strength can be increased because the target molecule can interact with the other walls of the pore without being affected by only one part of the pore [35]. Therefore, the membrane performance can be improved by incorporating the polymer-infiltrated MOF-808 into the polymer matrix [36].”
Comments #2
How does the MOF itself compares to the polymer/membrane? Does the MOF have the same selectivity? If the MOF was deposited as a film in the substrate how would it compare with the mixed-matrix described in this work?
Response #2
A better way to analyze the effect of MOF on gas separation is to obtain the gas separation performance of the pristine MOF membrane. Unfortunately, however, it is difficult to fabricate pristine defect-free MOF membranes on the Psf support without polymer matrix because MOFs are highly crystalline and rigid. It is very challenging to eliminate the defects between the MOF particles especially on a flexible polymer substrate. As a result, MOF membranes in polymer-based thin film composites are generally defective and exhibit very low selectivity. Considering the deterioration of gas separation performance upon excessive MOF loading, the higher the proportion of MOF particles, the lower the gas separation performance. Assuming an ideal MMM, the Maxwell model can be used to theoretically extrapolate the gas separation performance of pristine MOF-808, but the application of the model to thin film composite membranes may be inappropriate and may therefore have a large margin of error.

Round 2
Reviewer 2 Report
Response #1
It has been well addressed.
Response #2
The mentioned experiment has not been provided. The reviewer believes that the experiment to be supplemented is necessary to evaluate the adsorption selectivity. The author should try and take appropriate method to overcome the problems they may encounter, instead of thinking it is difficult and not doing it.
Author Response
Reviewer #2
The mentioned experiment has not been provided. The reviewer believes that the experiment to be supplemented is necessary to evaluate the adsorption selectivity. The author should try and take appropriate method to overcome the problems they may encounter, instead of thinking it is difficult and not doing it.
Response #2-1
First, we added Figure 3c to show the enhanced adsorption selectivity of PBE@MOF-808 compared to neat MOF-808. Second, we conducted two different experiments to prepare MOF membranes as suggested. First, a MOF thin-film membrane was prepared by the conventional solution-casting method used in our experiments (denoted as MOF-sol). A solution consisting of MOF dispersion without a polymer was casted on a PTMSP-coated polysulfone (Psf) support. As shown in Table 1, it showed very poor gas separation performance with a CO2 permeance of 12,363 GPU and CO2/N2 selectivity of 1.88. The performance was not much different from that of PTMSP-coated Psf (CO2 permeance 15,760 GPU, CO2/N2 selectivity 1.44). As expected, the gases freely permeated without selectivity through the defect between the MOFs. To overcome this problem, we tried to grow MOF-808 on the surface of Psf by in situ hydrothermal method (denoted as MOF@Psf), according to the previously reported procedure with a modification [Ma, K.; Islamoglu, T.; Chen, Z.; Li, P.; Wasson, M.C.; Chen, Y.; Wang, Y.; Peterson, G.W.; Xin, J.H.; Farha, O.K. Scalable and Template-Free Aqueous Synthesis of Zirconium-Based Metal–Organic Framework Coating on Textile Fiber. JACS 2019, 141, 15626-15633, doi:10.1021/jacs.9b07301.]. We used formic acid instead of trifluoroacetic acid (TFA) as a modulator since TFA is a very strong acid that can degrade Psf polymeric support. The membrane prepared with the hydrothermal method had lower gas permeance compared to neat Psf (CO2 permeance 20,850 GPU, CO2/N2 selectivity 0.94) by the growth of MOF-808 on the surface. Nonetheless, the gas selectivity was not improved, indicating that the selective MOF-808 membrane layer was not successfully fabricated. We have tried to fabricate a dense MOF-808 membrane through various methods, however, it was difficult to fabricate it as a thin-film on a polymeric support with selectivity.
Table 1. Gas separation performance of the pristine MOF-808 membranes within different preparation methods.
|
Membrane |
Fabrication method |
CO2 permeance (GPU) |
N2 permeance (GPU) |
CO2/N2 selectivity |
|
MOF-sol |
Solution-casting method |
12,363 |
6,569 |
1.88 |
|
MOF@Psf |
In situ hydrothermal growth on Psf |
9,402 |
9,796 |
0.96 |
On page 9, we have added following description.
“A MOF-808 membrane without the PBE copolymer was also prepared on a PTMSP-coated Psf support using the same method. The neat MOF-808 membrane showed poor gas separation performance with a CO2 permeance of 12,000 GPU and CO2/N2 selectivity of 1.9, indicating the importance of the PBE copolymer matrix to minimize the structural defects.”
Round 3
Reviewer 2 Report
The authors have addressed all of the issues and the manuscript can be accepted now.